behaviour/hybrid computing/complexity

social network, polarization, echo chamber, controlled experiment, privacy protection

**Author for correspondence:**
Yong Min
e-mail: myong@zjut.edu.cn

# Endogenetic structure of filter bubble in social networks

Yong Min[1], Tingjun Jiang[1], Cheng Jin[2], Qu Li[1] and Xiaogang Jin[3]

[1]College of Computer Science and Technology, Zhejiang University of Technology, Hangzhou, People's Republic of China
[2]Tencent Corporation, Shenzhen, People's Republic of China
[3]Zhejiang University, Hangzhou, People's Republic of China

YM, 0000-0002-9387-3921

The filter bubble is an intermediate structure to provoke polarization and echo chambers in social networks, and it has become one of today's most urgent issues for social media. Previous studies usually equated filter bubbles with community structures and emphasized this exogenous isolation effect, but there is a lack of full discussion of the internal organization of filter bubbles. Here, we design an experiment for analysing filter bubbles taking advantage of social bots. We deployed 128 bots to Weibo (the largest microblogging network in China), and each bot consumed a specific topic (entertainment or sci-tech) and ran for at least two months. In total, we recorded about 1.3 million messages exposed to these bots and their social networks. By analysing the text received by the bots and motifs in their social networks, we found that a filter bubble is not only a dense community of users with the same preferences but also presents an endogenetic unidirectional star-like structure. The structure could spontaneously exclude non-preferred information and cause polarization. Moreover, our work proved that the felicitous use of artificial intelligence technology could provide a useful experimental approach that combines privacy protection and controllability in studying social media.

## 1. Introduction

With the growing popularity of social media, especially microblogging platforms, the way people obtain information and form opinions has undergone substantial change. A recent study found that social media has become the primary source for over 60% of users to obtain news [1]. These users are selectively exposed to more personalized information, which is considered to limit the diversity of content they consume and give rise to filter bubbles [2–4]. News consumption on social media has been extensively studied to determine what factors lead to

polarization [5,6]. Recent works suggest that confirmation bias or selective exposure plays a significant role in online social dynamics [5,7]. That is, online users tend to select messages or information sources supporting their existing beliefs or cohering with their preferences and hence to form filter bubbles [4]. However, the polarization process, especially the features of filter bubbles, still needs further clarification.

In collective and individual levels, the term 'polarization' has two possible meanings. One is that like-minded people form exclusive clubs sharing similar opinions and ignoring dissenting views, i.e. opinion polarization [5]; the other is that an individual is exposed to less diverse content and is limited by a narrow set of information, i.e. information polarization [6]. For this paper, we focus on information polarization on social media.

The formation mechanism of polarization and the method of restraining polarization are the most pressing issues in the field of information polarization [2,4,7,8]. Many studies have suggested that selective exposure, both self-selection, and pre-selection, limits people's exposure to diverse content (i.e. filter bubble) and increases polarization [9]. Self-selection is the tendency for users to consume content or build new relationships that confirm their existing beliefs and preferences. Pre-selection depends on computer algorithms to personalize content for users without any conscious user choice. The present research generally agrees that self-selection is the primary cause of information polarization [7].

In the chain from selective exposure to information polarization, the formation and evolution of filter bubbles play as critical a role as the amplifier [3,10]. Therefore, quantifying and measuring filter bubbles has become a central issue in studying polarization [11,12]. From the perspective of opinion polarization, filter bubbles are usually considered equivalent to the community structure (densely connected internally) and are measured by modularity coefficient or community boundaries [10,13]. However, the community structure reflects more of a relationship between a filter bubble and the external network, rather than the internal organization of the filter bubble. By analysing news consumption on Facebook, the internal structural features of filter bubbles have been discovered, for example, users in filter bubbles are usually only concerned with a few information sources [6]. However, the details of filter bubbles still need further research.

At present, research about online social networks relies mainly on observational methods [6,7], which means that researchers cannot intervene in the study object but can only process and analyse the passively acquired data. However, the big open data contains great noise [14] and the risk of revealing privacy [15]. To overcome the shortcomings of a purely observational study, some researchers have ingeniously adopted the method of natural experiments or quasi-experiments to make comparative analyses and causal inferences from the available datasets [14,16–18]. Nevertheless, the uncontrolled approach limits the freedom of research. Recently, controlled experiments on social networks have deepened our understanding about information sharing and diffusion, behaviour spreading, voting and political mobilization, and cooperation [14,19–23]. Moreover, compared to the big data approach, controlled experiments can effectively control the impact factors, resulting in a causal relationship using a relatively small set of samples.

Performing experiments on real-world social networks, while actively promoting innovation of sociology and communication studies, also brings about some serious problems [24]. For instance, the political voter mobilization experiment conducted by Robert *et al.* on Facebook was sharply criticized by a considerable number of scholars, who argued that scientific research should not interfere with national politics or users' voting behaviour [22]. Therefore, the inappropriate design of actual large-scale social networks is likely to cause social injustice or infringe on the privacy of users.

In this paper, we report an experimental approach for studying the polarization process by using social bots [25]. We deploy 128 social bots to Weibo (NASDAQ: WB, the most famous Chinese Twitter-like service, with 430 million active monthly users). By analysing the data generated by these bots, we try to clarify the route from selective exposure to polarization, especially the structure and effect of filter bubbles. Most importantly, our approach is limited to using the data generated by the bots themselves, without any data about the actual person; therefore it is a privacy-friendly approach.

## 2. Method

### 2.1. Social bot design

A social bot is a computer algorithm that automatically interacts with the social network environment [25]. Social bots are generally considered to be harmful, although some of them are benign and, in principle, innocuous or even helpful. Therefore, social bots are often the subject of research that needs

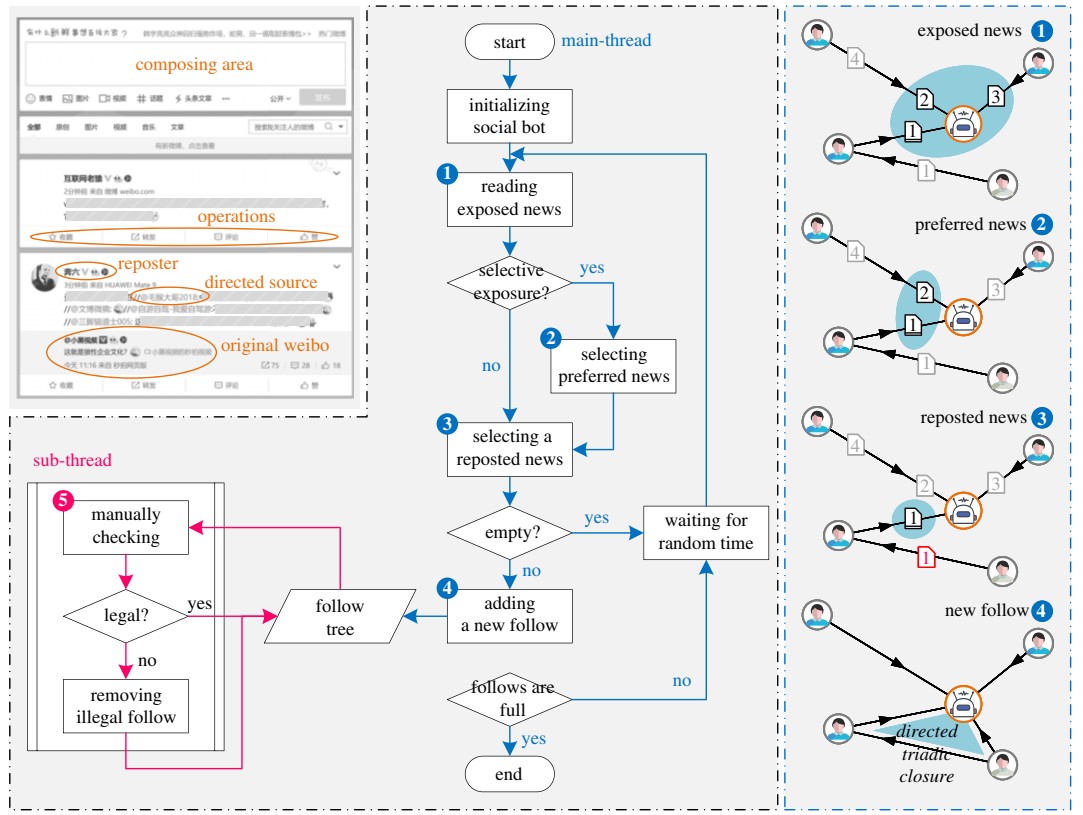

**Figure 1.** Design of neutral social bot. Based on the operating interface of Weibo (left), we use the flowchart (middle) and the schematic diagram (right) to illustrate the main workflow of the social robot, including the automated process (1–4) and the manual assistant process (5).

to be eliminated [26], but researchers have yet to recognize their potential value as a powerful tool in social network analysis [25,27]. The social bot in our experiment is designed to imitate similarity-based relationship formation, which reflects the selective exposure of information and relationships depending on one's preferred topics.

The social bot is based on two well-known hypotheses. The first is triadic closure [28]. Triadic closure suggests that among three people, A, B, and C, if a link exists between A and B, and A and C, then there is a high probability of a link between B and C. The hypothesis reflects the tendency of a friend of a friend to become a friend and is a useful simplification of reality that can be used to understand and predict the evolution of social networks. Because the relationships embedded in Weibo should be modelled by a directed graph, the bots use directed triadic closure to expand their followings (figure 1) [29]. The second hypothesis is homophily [5], i.e. people who are similar have a higher chance of becoming connected. In online social networks, for example, an individual creating her new homepage tends to link it to sites that are close to her interests (i.e. preferences).

Based on the two hypotheses, the workflow of the bots includes five steps (figure 1). (1) Initially, each bot is assigned 2 or 3 default followings, who mostly post or repost messages consistent with the topic of the bot. (2) A bot will periodically awaken from idleness at a uniformly random time interval. When the bot awakens, it can view the latest messages posted or reposted by its followings. As is well known, not all unread information can be exposed to users of social networks. A bot can assess only the latest 50 messages (i.e. the maximum amount in the one page on Weibo) after waking up. Please note that we exclude the influence of algorithmic ranking and recommendation systems on the information exposure by re-ranking all possible messages according to the descending order of posting time. (3) After viewing the exposed messages, the bot selects only the messages consistent with the topic. In this step, we first run the FastText text classification algorithm [30] to get preliminary classification results. To ensure the accuracy and correctness of classification, the results from the FastText algorithm were further voted by at least three experimenters. Although manual supervision is required, this algorithm can help us filter out a large number of inconsistent messages. (4) If there are reposted messages in selected messages, according to directed triadic closure, the bot randomly selects

a reposted message and follows the direct source of the reposted message. Please note that Weibo limits direct access to the information about the followings and followers of a user; thus, bots must find new followings through reposting behaviour. (5) If the following number reaches the upper limit, the bot stops running; otherwise, the bot becomes idle and waits to wake up again. The upper limit of followings for each bot is 150, according to the Dunbar number, which is a suggested cognitive limit to the number of people with whom one can maintain stable social relationships [31]. To avoid legal and moral hazards, the bots in the experiment do not produce, modify, or repost any information.

## 2.2. Experiment design

We are interested in exploring the following overarching question: how does selective exposure lead to polarization? Based on the above neutral social bots, we designed a controlled experiment to help us observe the news consumption and the evolution of personal social networks of the bots with a specific topic. For each bot, we mainly measured the following response variables:

V1: The probability that the bot is exposed to the preferred topic. For a given time $t$, the probability can be quantified by the preferred topic ratio ($R$), which is the ratio of the amount of information matching the topic to the total amount of information from 0 to $t$. Since the running of each bot is not completely synchronized, we normalize the time scale from 0 to 1, where $t = 0$ represents the initial state after being given the initial user, and $t = 1$ represents the time when the bot ends the running.

V2: The distribution of word frequency in the preferred topic. For a word $i$, the word frequency ($F$) is defined as the ratio of the number of messages that contain the word ($N_i$) to all exposed preferred messages of the bot ($N_t$):

$$F_i = \frac{N_i}{N_t}. \tag{2.1}$$

V3: The proportion of followings with the same preferred topic ($P$). The preference of followers is judged according to their nickname as well as the user tags and content of their microblogs.

V4: The structure of the personal social network ($G(n, m)$, where $n$ is the number of nodes and $m$ is the number of directed arcs) of a bot, including:

V4.1: in- and out-degree distribution.

V4.2: directed clustering coefficient ($C$) [32]:

$$C_x = \frac{T(x)}{\deg^{\text{tot}}(x)(\deg^{\text{tot}}(x) - 1) - 2\deg^{\leftrightarrow}(x)} \tag{2.2}$$

and

$$C = \overline{C_x} \text{ for all node } x, \tag{2.3}$$

where $T(x)$ is the number of directed triangles involved node $x$, $\deg^{\text{tot}}(x)$ is the sum of in-degree ($\deg^{\text{in}}$) and out-degree ($\deg^{\text{out}}$) of $x$ and $\deg^{\leftrightarrow}(x)$ is the reciprocal degree of $x$.

V4.3: Connection density ($D$):

$$D = \frac{m}{n(n-1)}. \tag{2.4}$$

V4.4: The dyadic and triadic motifs. Motifs are the special sub-structures indicating connecting patterns and functions in complex networks [33,34]. Motif has been applied to detect and measure controversy on Twitter [35]. In calculating motifs, we also consider the centrality (i.e. $\deg^{\text{in}}$) of nodes in personal social networks.

For V1 to V3, we can quantify the extent of the diversity of new consumption. For V4, we can detect the structure of filter bubbles for causing polarization.

We consider two most active 'soft' topics in Weibo: entertainment and science/technology (sci-tech). The entertainment topic includes celebrity gossip, fashion, movies, TV shows, music, etc. and the sci-tech topic involves nature, scientific theory, engineering, technological advances, digital products, the Internet, and so on. Compared with other topics, the two topics can be freely diffused in Weibo, have low overlap in content, and possess different user characteristics (gender, age, education level, etc.) [36,37]. For each topic, we designed two experimental treatments: topic group and random group. In

**Table 1.** An example of a table.

| standard | entertainment | sci-tech |
|---|---|---|
| accept | (1) celebrity gossip, fashion, movies, TV shows, and pop music; (2) explicitly containing the name, account or abbreviation of entertainers | (1) nature, scientific theory, engineering, technological advances, digital products, and Internet; (2) technical company and university |
| common reject | (1) commercial advertisement; (2) less than 5 Chinese characters | |
| specific reject | (1) ACG content (i.e. animation, comic, and digital game); (2) art, literature, and personal feeling; (3) simple lyrics and lines; (4) personal leisure activities | (1) financial or business report of technical or Internet company; (2) price of digital products; (3) military equipment; (4) daily skills; (5) constellations and divination; (6) weather forecast; (7) environmental conservation; (8) documentary with irrelevant content |

the topic group, bots select preferred content to expand their social networks, but in the random group, bots randomly choose new information sources in all exposed content disregarding the preferred topic. As a result, we have four experimental bot groups: topic environment group (EG), topic sci-tech group (STG), random entertainment group (REG), and random sci-tech group (RSTG). Each treatment contained 30–34 bots who operated on Weibo between 13 March and 28 September 2018. The initial followings come from a large enough user pool, which contained at least 100 candidates for each preference. For each topic, all bots in the topic and the random group have the same initial followings. The idle period of all bots was 2–4 h.

Compared with the conventional method, our design is characterized by using social bots as the agents to conduct experiments in real online social media. The approach can be seen as a Monte Carlo simulation in a real-world environment by adopting an artificial intelligence algorithm. These bots act entirely on predefined hypotheses and algorithms. Therefore, our experiment can be regarded as a type of randomized assignment.

## 2.3. Standard for preference classification

The most significant feature of the social bots is that they can autonomously identify whether information matches their preferred topic. Therefore, the criteria of text classification are fundamental. We defined a classification standard for the two topics in our experiment (table 1).

The classification criteria shown in table 1 were used to select initial followings, prepare the training corpus of the FastText classifier, manually verify, and result analysis. The unified standard can ensure consistency in the judgement of topic and the classification of text.

# 3. Results and discussion

## 3.1. Topic drives polarization

In social media, selective exposure of information source and information has been considered the primary mechanism of polarization. By comparing the preferred topic ratio in four experimental treatments, we can reveal the link between selective exposure and information polarization.

For two topics, since the initial users have been set as the users who mainly post or repost the corresponding topic, the initial ($t = 0$) preferred topic ratio can be as high as $\overline{R^0_{\mathrm{EG}}} = 76.77\%$ for EG and REG and $\overline{R^0_{\mathrm{STG}}} = 50.00\%$ for STG and RSTG on average (figure 2$a$). Because the topic group and the random group have the same initial followings, they also have the same initial $R$ for each topic. That is, most of the messages exposed to four groups are concentrated in the corresponding topics. Even so, the entertainment topic is more polarized than the sci-tech topic initially. The $\overline{R^0_{\mathrm{EG}}}$ is higher than 50% of $\overline{R^0_{\mathrm{STG}}}$ ($p < 0.01$, two-sided Mann–Whitney U test, 34 bots). Importantly, no matter how many different topics are exposed to them after a random wake-up, the bots in topic groups only

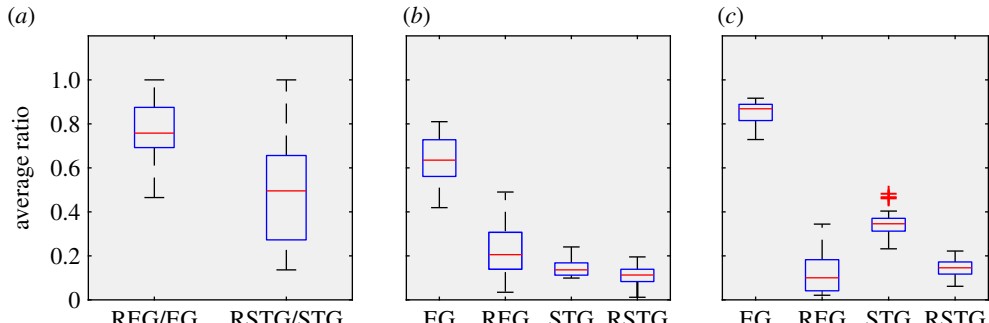

**Figure 2.** The polarization of exposed content and followings. (*a*) The preferred topic ratio in the initial state ($R^0$). Because the topic group and the random group have the same initial followings, they also have the same $R^0$ for each topic. (*b*) The preferred topic ratio in the final state $R^1$. (*c*) Proportion of followings with the same preferred topic ($P$) in four treatments.

select a message consistent with its preference and follow the direct source of the selected news. It means that the initial ($t = 0$) preferred topic ratio does not affect the primary experimental process of choosing messages and following users of the bots. Finally, the initial difference between EG and STG provides a baseline for our further analysis.

Compared to the initial state, we are more concerned about the diversity of messages consumed by social bots after their social networks have formed (figure 2*b*). First, at the end of the bots ($t = 1$), $\overline{R^1_{EG}}$ declines only slightly from $\overline{R^0_{EG}} = 76.77\%$ to 63.78%, but is significantly larger than $\overline{R^1_{REG}} = 19.23\%$. The results suggests that selective exposure is important but not sufficient for the polarization. By contrast, $R^1_{STG}$ and $R^1_{RSTG}$ is similar, but greatly less than $R^0_{STG}$ ($p < 0.01$, two-sided Mann–Whitney U test, 34 bots). The difference between the two topics suggests whether selective exposure causes polarization to be dependent on the topic.

The followings of a bot are the direct source of its news consumption. Our results showed that for the entertainment topic, the average proportion of nodes with the preferred topic $\overline{P_{EG}} = 85.32\%$ (up to 92.14% and down to 73.47%) (figure 2*c*); however, $\overline{P_{REG}}$ is only 15.27% without selective exposure. The large difference between $P_{EG}$ and $P_{REG}$ suggests that selective exposure can effectively filter the information source, making most of the information sources consistent with the preferred topic and thus resulting in a filter bubble. For the sci-tech topic, however, $\overline{P_{STG}} = 35.33\%$ (up to 47.86% and down to 23.12%), and $\overline{P_{RSTG}} = 14.51\%$ (figure 2*d*). Based on the information diffusion mechanism of Weibo and the design of social bots, a low proportion indicates that users preferring the sci-tech topic have a high possibility of following users liking other topics, and diverse users diffuse the sci-tech content. The overlap of topics can suppress the effect of selective exposure in forming filter bubbles.

The polarization is also reflected on the semantic level. In both EG and STG treatments, the word frequency of the preferred topic exposed to the social bots presents a power-law distribution (figure 3*a*). The power-law distribution ($y \sim x^{-\alpha}$) indicates that there are some dominant words with high word frequency in the text. Moreover, the dominance of a few words is more severe in EG than in STG ($\alpha = 2.23$ for EG and $\alpha = 4.74$ for STG). In the EG treatment, the maximum frequency of a word can be up to 0.4 on average, which means that the same word can be found in 40% of messages on average. However, in the STG treatment, the maximum word frequency is no more than 0.15 (figure 3*b*). Even among the top 10 words of EG, the first and second words are more dominant than other words. By analysing the top 10 high-frequency words of the two treatments, we find that the high-frequency words of EG are usually the names of certain entertainment celebrities and related specific words, while the high-frequency words of STG are some common words, such as 'China', 'America', 'technology', 'market' and 'company' (figure 3*c*).

## 3.2. The personal social networks

From the results in the previous section, we found that EG and STG treatments exhibit different levels of polarization. Therefore, by comparing the differences between EG and STG personal social networks, the structural characteristics of the filter bubbles can be found, which may be one of the mechanisms that cause polarization.

### 3.2.1. Statistical properties of networks

By roughly comparing the in- and out-degree distributions of EG and STG's personal social networks, we find that both present power-law distributions ($\alpha = 1.766$ for in-degrees and $\alpha = 1.847$ for out-degrees) and

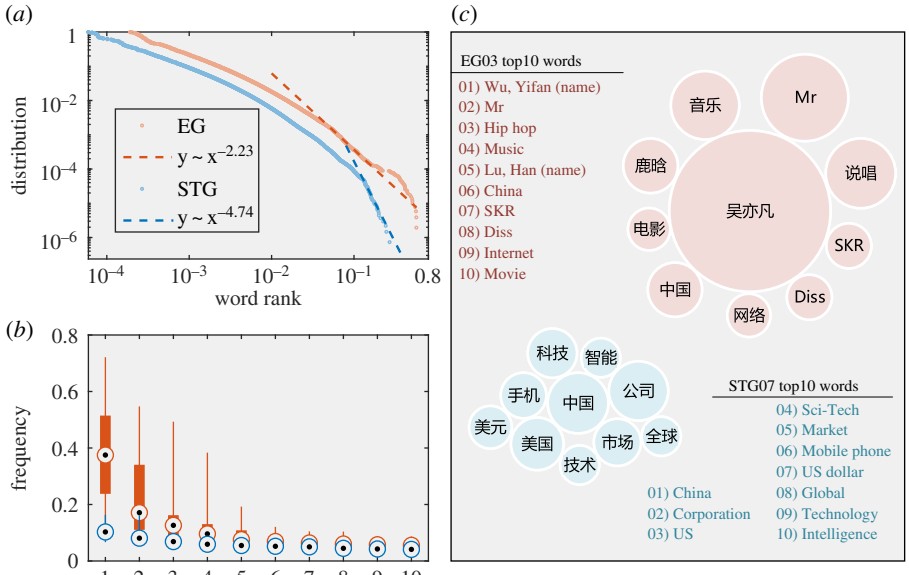

**Figure 3.** Semantic analysis of exposed contents. (*a*) Inverse cumulative distribution of word frequency *F* (log-log plot). The EG has a longer tail than the STG; i.e. EG has more higher-frequency words. (*b*) Box plots of the frequency of the top 10 words in both the EG and STG. (*c*) Demonstration of the top 10 words in exposing messages from EG 03 bot and STG 07 bot. The size of the bubble indicates *F*.

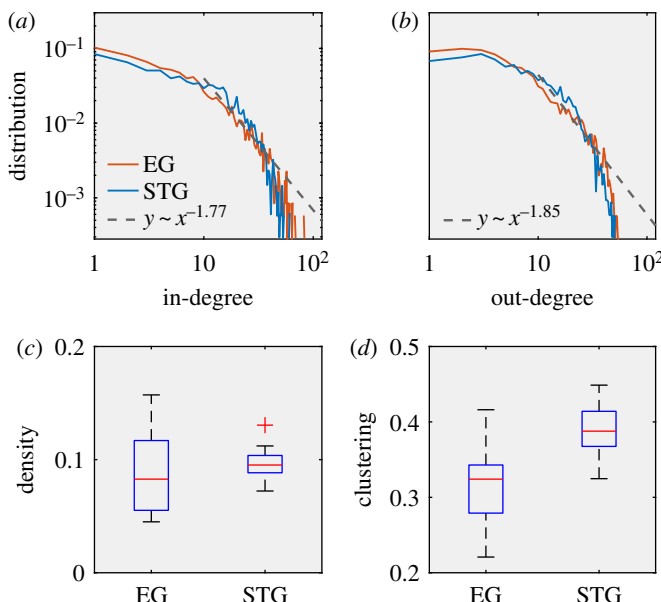

**Figure 4.** Structural features of personal social networks. (*a*) Mean of the inverse cumulative distribution of in-degree (log-log plot). (*b*) Mean of the inverse cumulative distribution of out-degree (log-log plot). (*c*) Connection density *D*. (*d*) Clustering coefficient *C*.

are almost identical (figure 4*a*,*b*). Moreover, the two groups of networks also have similar connection densities (figure 4*c*). Interestingly, figure 4*d* shows that the average clustering coefficient of STG networks is 35% higher than that of EG networks ($p < 0.01$, two-sided Mann–Whitney U Test, 34 networks). Given the low level of information polarization in STG, this is a counterintuitive result, as a dense community spreads diverse information.

### 3.2.2. Motifs

The previous counterintuitive phenomenon can be attributed to differences in the connection pattern of personal social networks. First, we consider all possible motifs between two nodes (i.e. followings of bots) in the networks. By considering the in-degree centrality of nodes (i.e. the number of followers of the node), there are five possible configurations, which are shown in figure 5*a*. Figure 5*b*

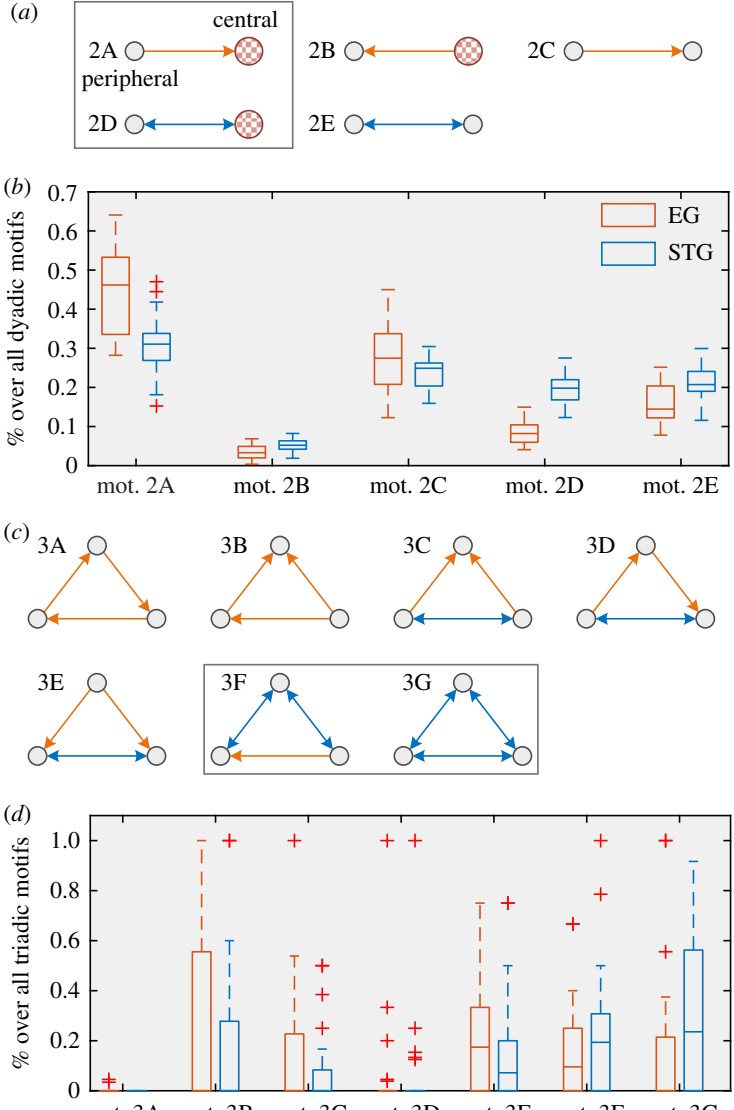

**Figure 5.** Motifs of personal social networks. For dyadic motifs, we distinguish between the central and the peripheral node. The boxes indicate the key motifs to distinguish filter bubbles.

shows the frequency distribution of dyadic motifs in EG and STG treatments. For two nodes of a directed arc, if the difference between centralities is larger than a threshold $T = 10$, we distinguish two nodes into a peripheral node and a central node, such as motif 2A and 2B in figure 5a. Note that motifs are mutually exclusive; therefore, a reciprocal arc will not be double-counted as two unidirectional arcs. The primary difference between EG and STG networks is the frequency of motif 2A and 2D. According to figure 5b, EG networks contain more non-reciprocal arcs from peripheral nodes to central nodes, but STG networks have more reciprocal arcs between peripheral and central nodes. Second, we also consider 3-node motifs, in particular closed triangles (figure 5c). Due to the high number of possible motifs, we do not distinguish centrality in calculating triadic motifs. Figure 5d shows that, compared with EG networks, STG networks contain more closed triangles with at least two reciprocal arcs.

The abundance of motifs 2D, 3F and 3G confirm the counterintuitive result derived from the statistical properties of the network, that is, the STG network is more clustering but less polarized. The form of motifs explains this result. In STG networks, if each peripheral node consumes only a small number of messages on other topics, the information has a relatively high probability of exposure to the central nodes. Consequently, the preference of central nodes may be affected, and broadcast the diverse messages to its followers, making various information more widely spread within the whole community.

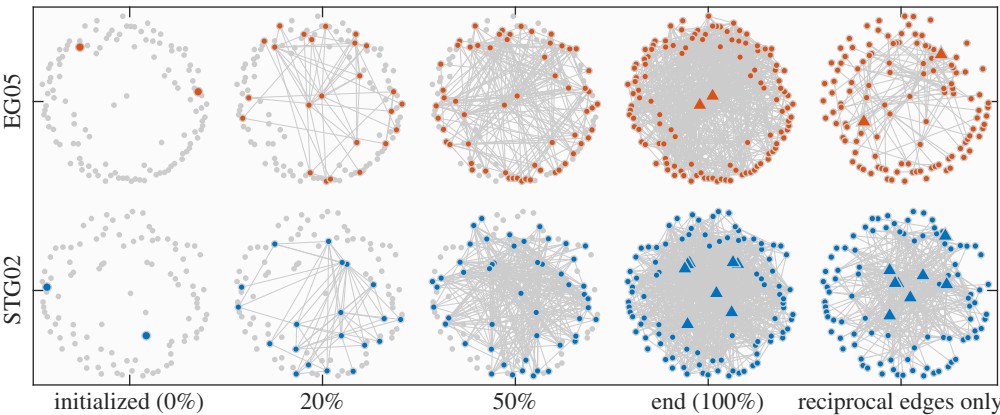

**Figure 6.** Visualization of the evolution of personal social networks. In this demonstration, Social bot 05 in EG (EG05) and bot 02 in STG (STG02) display a similar growth process from initialized two followings to the end of running. However, STG02 obtains more reciprocal edges than EG05. The triangles represent the nodes with high in-degree, that is, the corresponding users have a large number of followers. The visualization is based on the radial layout with in-degree centrality; thus the node with higher in-degree is closer to the centre of the plot.

### 3.2.3. Visualization

The visualization analysis also shows similar results. If all non-reciprocal arcs are removed, we find that the STG networks still have a high connection density, and the original central nodes are still maintained, while EG networks are the opposite, with the sparse connection, and the original central nodes degenerate into peripheral nodes (figure 6). Based on the above results, an EG network tends to be a unidirectional star-like structure with a few high-degree nodes, while an STG network tends to be a bidirectional clustering structure. The unidirectional star-like structure means selecting one or several central nodes to play the role of broadcasters to send information to all other nodes with few interactions between them. However, the diffusion path of this structure is mainly dependent on the selected central node. If the central nodes in the star-like structure produce only a narrow set of news, the others will be limited to a lower diversity of content. This selection effect of central nodes tends to enlarge the polarization of the content [38]. In the bidirectional clustering structure, all nodes are placed in a clustering group, and most connections are bidirectional. Those nodes can efficiently exchange information and realize a complementary effect with their friends with distinct preferences. The complementary effect can promote the diversity of information [38].

Furthermore, our results may confirm the basic structure and behaviour of the scientific community. The bidirectional scale-free structure of scientific collaboration networks has long been recognized [39,40]. Our results show that the scientific community forms a similar interactive centre-periphery structure, both in formal collaborations and in daily life (even in a unidirectional relationship-based social environments, e.g. microblogging). In Wu *et al.* [41], the researchers found an interactive pattern of large teams at the centre of the scientific community and small teams at the periphery of the scientific community. In the pattern, small teams make disruptive innovation, while large teams further develop and spread such innovation. The pattern has a similar connotation to the bidirectional structure we found. Peripheric users in the scientific community are responsible for providing novel information, while the central user is responsible for further disseminating the information. As a result, the coupling of the scale-free structure and bidirectional information dissemination forms a positive feedback loop that stimulates scientific innovation. At the same time, the bidirectional structure of the scientific community can promote communication between users and limit the spread of misinformation compared to the star-like structure of the entertainment community.

In sum, the controlled experiments based on neutral social bots help us to reveal the two basic structural characteristics of the filter bubbles in microblogging platforms. First, the filter bubble contains a high proportion of users preferring an exclusive topic. Second, filter bubbles have a unidirectional star-like social structure, in which the central nodes play only the role of the information source, and rarely receives information from other nodes. The combination of two characteristics leads to the emergence of information polarization.

# 4. Conclusion

In this paper, we report a new controlled experiment approach to study polarization in social media. This method can be seen as a real-time Monte Carlo simulation on real-world social networks using the computer and artificial intelligence technology. (1) Compared with the experimental method based on volunteers [42], social bots use artificial intelligence technologies (e.g. natural language processing) to simulate specific actions (e.g. selective exposure). The approach not only improves the freedom of the experimenters but also reduces the cost of the experiments on real-world social networks. Unlike volunteers, the behaviour of social bots can be effectively controlled; therefore, the consistency and randomness of experiment processes can be ensured, such as activity frequency and content judgement criteria. (2) Compared with the experimental method based on artificial social networks [19,20], social bots can work directly on real-world social networks and acquire timely feedback with the instantaneous online environment. (3) Compared with the experimental method of directly altering the real-world social network [22], this method can limit the interference to real-world social networks to a lower level. In our experiment, all social bots we deploy have no direct interaction with the existing users except following on them. The bots do not produce, forward, or modify any news content, and all data collected is directly exposed to them and publicly accessible. All network connections are restricted within bots and their direct followings, and we do not use the entire social relationship of all followings. That is, our experiment does not interfere with users' behaviour or disturb information dissemination, and all the data used is publicly visible. Therefore, the approach avoids the risk of violating the privacy of users. However, limited by the current technical conditions, this method can intelligently simulate only relatively simple actions. How to simulate complex user behaviours (especially initiative feedback such as comment and chatting) is still a technical challenge.

Taking advantage of the approach, we can compare the consumption of different topics. Previous studies focused on the dissemination of different opinions in response to particular content, such as political news [6,10,43,44]. In this case, users were actually in the same media environment, and the differences in topics could be ignored. Second, the rise of pre-selection technologies (e.g. recommendation systems) makes people more attentive to the comparison of pre-selection and self-selection than to the impact of the media environment [7]. However, our research uses the same benchmark to compare the consumption of different news topics, thus revealing the internal structure of filter bubbles in microblogging platforms. We found that the inside of the filter bubble exhibits a power-law distribution (both in- and out-degree), which is consistent with previous research results, that is, users in the filter bubble usually pay attention to a few sources of information. For the directed graph structure of microblogs, we find that the reciprocal link between the central nodes and the peripheral nodes plays an important role in the polarization process. For filter bubbles, peripheral nodes usually unidirectionally follow the central node, while for non-polarized communities, the central nodes tend to interact with others bidirectionally. The difference can be quantified by particular dyadic and triadic motifs (figure 5). The result provides an alternative way to measure and minimize polarization in social media.

Data accessibility. The Java code to repeat the experiment and the Weibo data involved in this paper can be found in the electronic supplementary material or downloaded from our website: www.socialbot.top.
Authors' contributions. Y.M. designed the experiment, coded the social bots, and drafted the manuscript; T.J. carried out the statistical analyses; C.J. participated in experiment design and critically revised the manuscript; Q.L. and X.J. helped draft the manuscript. All authors gave final approval for publication and agree to be held accountable for the work performed therein.
Competing interests. We declare we have no competing interest.
Funding. This work was supported by the National Natural Science Foundation of China (grant nos. 71303217, 31370354 and 31270377) and the Zhejiang Provincial Natural Science Foundation of China (grant nos. LY17G030030, LGF18D010001 and LGF18D010002).
Acknowledgements. We thank Prof. Jie Chang and Prof. Ying Ge (Zhejiang University) for their guidance on the diversity theory. We thank Haidan Yang for her introduction to the theory of communication. We thank Aizhu Liu, Chenyi Fang, Conger Yuan, Fan Li, Hao Li, Hao Wu, Haochen Hou, Hengji Wang, Jian Zhou, Jiaye Zhang, Jinmeng Wang, Junhao Xu, Lingjian Jin, Longzhong Lu, Lu Chen, Luchen Zhang, Qiuhai Zheng, Qiuya Ji, Renyuan Yao, Ruonan Zhang, Shang Gao, Shicong Han, Songyi Huang, Ting Xu, Wei Fang, Wei Zhang, Xingfan Zhang, Yijing Wang, Yingjie Feng, Yinting Chen, Yiqi Ning, Yujie Bao, Yuying Zhou, Zheyu Li, Ziyu Liu for their contribution to the manual text classification.

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
