## [Reviewer comments · Royal Society Open Science]

Review History

RSOS-190868.R0 (Original submission)

Review form: Reviewer 1

Is the manuscript scientifically sound in its present form?

Yes

Are the interpretations and conclusions justified by the results?

No

Is the language acceptable?

Yes

Is it clear how to access all supporting data?

Yes

Do you have any ethical concerns with this paper?

No

Have you any concerns about statistical analyses in this paper?

I do not feel qualified to assess the statistics

Recommendation?

Accept with minor revision (please list in comments)

Comments to the Author(s)

This paper collects information feeds by deploying 128 bots in Sina Weibo (chinese equivalent of Twitter). By analysing the text and network motifs in their social networks, the paper makes two discoveries/claims:

1. that filter bubbles have an "endogenetic" unidirectional star-like structure.
2. that the use of bots and AI could combines privacy protection and provides controllability in social media studies.

Related work is well introduced and placed in context.

The paper itself is reasonably interesting, but I need more convincing arguments that it is valid, as explained below.

Problems:

The main issues I have with the paper are two fold:

1. The social bots not in the Random groups are designed to selectively enrich their connections within topic. So of course, we are bound to see "polarization" as seen in Section 3(a). In fact what is surprising is that there is not much polarization difference between STG and RSTG. To understand this difference, a complicated (and not entirely convincing) network analysis is performed. Instead, I imagine this arises just as a straightforward side effect of how much topic specialisation exists among the seed users that are followed, and in general all users in STG. What I imagine is that the users who are being followed by the bots in STG are users who tweet about science but also other topics ($R^0_{\{STG\}} = 0.5$). On the other hand, EG bots follow users who only tweet about entertainment and nothing else $R^0_{\{EG\}} = 76.77\%$. Therefore, when looking at the previous 50 messages after a random wakeup, and choosing users to follow, STG bots may find other topics, and users who tweet about those topics. In contrast, EG bots simply find other users also tweeting mainly about EG.
2. There are only two topics tested. On one topic (EG) it is found (page 7, line 16) that selective exposure is necessary for polarization. On another topic (STG) this effect is not found. With this, the natural conclusion is that we simply cannot generalise, and that perhaps there is no universal truth being found - just some artefact of these specific topics.

A third "cosmetic" issue is in the framing the claimed method - ie contribution 2: It is not clear why authors need to deploy "agents" or social bots. The same could have been achieved by simulating the following and reading policies of different kinds of users. At the heart is just the policy which is being followed, whether that is a bot, or a human. The fact that a bot has been used is "implementation detail". I am also not sure how it avoids the "invasion of privacy" (Page 12, line 25).

Page 5, line 26: "results from the algorithm are further voted..." -> was every step confirmed by three experimenters? How many steps did the bots take in total?

Related to this, in line 32, it is stated that bots stop after reaching an upper limit of number of followers. It is not clarified until next page that this limit is based on Dunbar's number. Please state here.

Page 7, line 29–30: "Compared with other topics, the two topics can be freely diffused in Weibo, have low overlap in content, and possess different user characteristics (gender, age, education level, etc.)."

This needs a reference to confirm they have different user characteristics.

Fig 2: Please do not use a pie chart. There are several statistical issues identified with pie charts (<https://www.data-to-viz.com/caveat/pie.html>)

I dont quite follow the motifs explanation in section 3b.ii - it concludes saying STG networks contain more closed triangles with at least two reciprocal arcs. Isnt this to be expected given that dyadic analysis already found more reciprocity? Also, a couple of sentences are needed on how exactly this motif analysis explains the counterintuitive result noted in Page 10 line 1.

Review form: Reviewer 2 (Marko Curkovic)

Is the manuscript scientifically sound in its present form?

Yes

Are the interpretations and conclusions justified by the results?

Yes

Is the language acceptable?

Yes

Is it clear how to access all supporting data?

Yes

Do you have any ethical concerns with this paper?

Yes

Have you any concerns about statistical analyses in this paper?

No

Recommendation?

Accept with minor revision (please list in comments)

Comments to the Author(s)

The manuscript is a well written piece, that has a significant contribution to the field, and it is methodologically sound. The main strength of the study is that it is the first study that is comprehensively evaluating the use of social bot and artificial intelligence (although supervised) in researching inherent structure of filter bubbles occurring in online social spaces (in this case Weibo microblogging network). It is also methodologically sound as author used four distinct groups: topic and random group, whereas researching filter bubble polarization processes/effects and network structures on two distinct topics: entertainment and sci-tech.

I could imagine that authors had difficulties singling-out and discussing few of many interesting findings their study yielded.

So, as results suggest, there is indeed difference between polarization potential and network structure between two given topics: entertainment and sci-tech. Network emerging within

entertainment topic is more “unidirectional, star-like”, whereas sci-tech “bidirectional”. This is valuable finding and could be reflective to the topic/information per se – entertainment and sci-tech. Sci-tech network/community could be more “inductive” (being reflective and incorporating many diverse sources. And therefore, spreading diverse information), while entertainment “deductive” (being responsive to less diverse sources, ones dictated by few central agents). In other words, there is indeed difference in the content of information, and how it drives all the examined processes: polarization, self/other-reliance, network structure (although density seems to be similar), etc.

So, I am a bit confused when conclusions of the manuscript (seen also in the abstract of the manuscript) are mostly based on findings that emerged around entertainment topic. Moreover, as previously it has been suggested that similar issues indeed might occur in the “official” scientific community, and that similar processes might have impact on scientific evidence (even once considered as most reliable such as systematic reviews and meta-analyses). So, I would prefer to see more detailed discussion (within Discussion section) regarding the differences found within these two topics.

Technical issues:

„In collective and individual levels, the term ‘polarization’ has two possible meanings.” - misspelling

In conclusion section points from (1) to (5) needs to be rearranged, as currently it is difficult to follow and there are some issues with spelling (for example after point 4).

Supplementary files can be easily accessed as mentioned within the manuscript, but website: www.socialbot.top is in Chinese.

Regarding ethics:

The authors initially stated that “This article does not present research with ethical considerations.”

I would personally prefer that ethical committee have approved this research protocol, or that it has been pre-registered (there is no mention of such an approval nor pre-registration in the text of the manuscript).

This study is to a certain point proof-of-concept study, as authors demonstrate that it is feasible and secure to use social bots and artificial intelligence to research social and communication patterns in online space (certainly pressing issues).

However, no matter how safe this method is, it should be regulated, approved and supervised by some form of “external” agent, as it does involve human actors (although passively and without immediate threat – there are indeed no threats to privacy as most difficult issue in such a research). Although social bots are passive in terms of creating and forwarding content their involvement does involve certain degree of “deception” to human user on the other end of the interaction.

Finally, this research experiment does occur in online space that is supervised per se (by the owner) and using social bots could be against owner policies. One can assume that microblogging networks are using similar mechanisms themselves in order to make their products more successful/efficient.

However, as study is unique, I would not consider these issues as obstacle to publication, but some would like to see these issues discussed in more details in the manuscript. Especially as authors are suggesting this as valid and reliable method for such and similar research. So, then, we must imagine ourselves in future landscape where many similar studies are being simultaneously conducted, and authors recommendations need to address this.

Decision letter (RSOS-190868.R0)

16-Aug-2019

Dear Dr Min,

The editors assigned to your paper ("Endogenetic structure of filter bubble in social networks") have now received comments from reviewers. We would like you to revise your paper in accordance with the referee and Associate Editor suggestions which can be found below (not including confidential reports to the Editor). Please note this decision does not guarantee eventual acceptance.

In particular, the Editors draw your attention to the comments of the second reviewer regarding the ethical approvals required for the study -- please ensure you respond to and act upon these requirements from the reviewer.

Please submit a copy of your revised paper before 08-Sep-2019. Please note that the revision deadline will expire at 00.00am on this date. If we do not hear from you within this time then it will be assumed that the paper has been withdrawn. In exceptional circumstances, extensions may be possible if agreed with the Editorial Office in advance. We do not allow multiple rounds of revision so we urge you to make every effort to fully address all of the comments at this stage. If deemed necessary by the Editors, your manuscript will be sent back to one or more of the original reviewers for assessment. If the original reviewers are not available, we may invite new reviewers.

- Data accessibility

It is a condition of publication that all supporting data are made available either as supplementary information or preferably in a suitable permanent repository. The data accessibility section should state where the article's supporting data can be accessed. This section should also include details, where possible of where to access other relevant research materials such as statistical tools, protocols, software etc can be accessed. If the data have been deposited in

an external repository this section should list the database, accession number and link to the DOI for all data from the article that have been made publicly available. Data sets that have been deposited in an external repository and have a DOI should also be appropriately cited in the manuscript and included in the reference list.

<http://datadryad.org/submit?journalID=RSOS&manu=RSOS-190868>

- **Competing interests**

- **Authors' contributions**

- **Acknowledgements**

- **Funding statement**

Kind regards,

Andrew Dunn

on behalf of Dr Cecilia Mascolo (Associate Editor) and Marta Kwiatkowska (Subject Editor)
openscience@royalsociety.org

Associate Editor's comments (Dr Cecilia Mascolo):

Associate Editor: 1

Comments to the Author:

There are a few minor issues left which can be fixed with presentation adjustments. One aspect however to be discussed is related to ethics: one of the reviewers questions your choice of not putting this work through an ethics committee even just an overseeing body and we think this is probably best assessed by them before publication (there are ethics and legal aspects to be considered and possibly discussed in the paper).

Comments to Author:

Reviewers' Comments to Author:

Reviewer: 1

Comments to the Author(s)

This paper collects information feeds by deploying 128 bots in Sina Weibo (chinese equivalent of Twitter). By analysing the text and network motifs in their social networks, the paper makes two discoveries/claims:

1. that filter bubbles have an "endogenetic" unidirectional star-like structure.
2. that the use of bots and AI could combines privacy protection and provides controllability in social media studies.

Related work is well introduced and placed in context.

The paper itself is reasonably interesting, but I need more convincing arguments that it is valid, as explained below.

Problems:

The main issues I have with the paper are two fold:

1. The social bots not in the Random groups are designed to selectively enrich their connections within topic. So of course, we are bound to see "polarization" as seen in Section 3(a). In fact what is surprising is that there is not much polarization difference between STG and RSTG. To understand this difference, a complicated (and not entirely convincing) network analysis is performed. Instead, I imagine this arises just as a straightforward side effect of how much topic specialisation exists among the seed users that are followed, and in general all users in STG. What I imagine is that the users who are being followed by the bots in STG are users who tweet about science but also other topics ($R^0_{\{STG\}} = 0.5$). On the other hand, EG bots follow users who only tweet about entertainment and nothing else $R^0_{\{EG\}} = 76.77\%$. Therefore, when looking at the previous 50 messages after a random wakeup, and choosing users to follow, STG bots may find other topics, and users who tweet about those topics. In contrast, EG bots simply find other users also tweeting mainly about EG.
2. There are only two topics tested. On one topic (EG) it is found (page 7, line 16) that selective exposure is necessary for polarization. On another topic (STG) this effect is not found. With this, the natural conclusion is that we simply cannot generalise, and that perhaps there is no universal truth being found - just some artefact of these specific topics.

A third "cosmetic" issue is in the framing the claimed method - ie contribution 2: It is not clear why authors need to deploy "agents" or social bots. The same could have been achieved by simulating the following and reading policies of different kinds of users. At the heart is just the policy which is being followed, whether that is a bot, or a human. The fact that a bot has been used is "implementation detail". I am also not sure how it avoids the "invasion of privacy" (Page 12, line 25).

Page 5, line 26: “results from the algorithm are further voted...” → was every step confirmed by three experimenters? How many steps did the bots take in total?

Related to this, in line 32, it is stated that bots stop after reaching an upper limit of number of followers. It is not clarified until next page that this limit is based on Dunbar’s number. Please state here.

Page 7, line 29–30: "Compared with other topics, the two topics can be freely diffused in Weibo, have low overlap in content, and possess different user characteristics (gender, age, education level, etc.)."

This needs a reference to confirm they have different user characteristics.

Fig 2: Please do not use a pie chart. There are several statistical issues identified with pie charts (<https://www.data-to-viz.com/caveat/pie.html>)

I don't quite follow the motifs explanation in section 3b.ii - it concludes saying STG networks contain more closed triangles with at least two reciprocal arcs. Isn't this to be expected given that dyadic analysis already found more reciprocity? Also, a couple of sentences are needed on how exactly this motif analysis explains the counterintuitive result noted in Page 10 line 1.

Reviewer: 2

Comments to the Author(s)

The manuscript is a well written piece, that has a significant contribution to the field, and it is methodologically sound. The main strength of the study is that it is the first study that is comprehensively evaluating the use of social bot and artificial intelligence (although supervised) in researching inherent structure of filter bubbles occurring in online social spaces (in this case Weibo microblogging network). It is also methodologically sound as author used four distinct groups: topic and random group, whereas researching filter bubble polarization processes/effects and network structures on two distinct topics: entertainment and sci-tech.

I could imagine that authors had difficulties singling-out and discussing few of many interesting findings their study yielded.

So, as results suggest, there is indeed difference between polarization potential and network structure between two given topics: entertainment and sci-tech. Network emerging within entertainment topic is more “unidirectional, star-like”, whereas sci-tech “bidirectional”. This is valuable finding and could be reflective to the topic/information per se – entertainment and sci-tech. Sci-tech network/community could be more “inductive” (being reflective and incorporating many diverse sources. And therefore, spreading diverse information), while entertainment “deductive” (being responsive to less diverse sources, ones dictated by few central agents). In other words, there is indeed difference in the content of information, and how it drives all the examined processes: polarization, self/other-reliance, network structure (although density seems to be similar), etc.

So, I am a bit confused when conclusions of the manuscript (seen also in the abstract of the manuscript) are mostly based on findings that emerged around entertainment topic. Moreover, as previously it has been suggested that similar issues indeed might occur in the “official” scientific community, and that similar processes might have impact on scientific evidence (even once considered as most reliable such as systematic reviews and meta-analyses).

So, I would prefer to see more detailed discussion (within Discussion section) regarding the differences found within these two topics.

Technical issues:

„In collective and individual levels, the term „a ̃rpolarization ˆa’s has two possible meanings.“ - misspelling

In conclusion section points from (1) to (5) needs to be rearranged, as currently it is difficult to follow and there are some issues with spelling (for example after point 4).

Supplementary files can be easily accessed as mentioned within the manuscript, but website: www.socialbot.top is in Chinese.

Regarding ethics:

The authors initially stated that “This article does not present research with ethical considerations. “

I would personally prefer that ethical committee have approved this research protocol, or that it has been pre-registered (there is no mention of such an approval nor pre-registration in the text of the manuscript).

This study is to a certain point proof-of-concept study, as authors demonstrate that it is feasible and secure to use social bots and artificial intelligence to research social and communication patterns in online space (certainly pressing issues).

However, no matter how safe this method is, it should be regulated, approved and supervised by some form of “external” agent, as it does involve human actors (although passively and without immediate threat – there are indeed no threats to privacy as most difficult issue in such a research). Although social bots are passive in terms of creating and forwarding content their involvement does involve certain degree of “deception” to human user on the other end of the interaction.

Finally, this research experiment does occur in online space that is supervised per se (by the owner) and using social bots could be against owner policies. One can assume that microblogging networks are using similar mechanisms themselves in order to make their products more successful/efficient.

However, as study is unique, I would not consider these issues as obstacle to publication, but some would like to see these issues discussed in more details in the manuscript. Especially as authors are suggesting this as valid and reliable method for such and similar research. So, then, we must imagine ourselves in future landscape where many similar studies are being simultaneously conducted, and authors recommendations need to address this.

Author's Response to Decision Letter for (RSOS-190868.R0)

See Appendices A & B.

Decision letter (RSOS-190868.R1)

07-Oct-2019

Dear Dr Min,

I am pleased to inform you that your manuscript entitled "Endogenetic structure of filter bubble in social networks" is now accepted for publication in Royal Society Open Science.

on behalf of Dr Cecilia Mascolo (Associate Editor) and Marta Kwiatkowska (Subject Editor)
openscience@royalsociety.org

Associate Editor Comments to Author (Dr Cecilia Mascolo):
Associate Editor

Comments to the Author:

The review answers all the reviewers comments including the ethics concerns (a letter from the board approval is attached). I am also satisfied that these concerns are minor and well explained now in the paper.

Follow Royal Society Publishing on Twitter: [@RSocPublishing](https://twitter.com/RSocPublishing)

Appendix A

Response Letter

Dear Editors and Reviewers,

Thanks very much to the editors and reviewers for your valuable comments on this manuscript. In this response letter, we will explain our revision for editors and reviewers' comments one by one with underline blue text.

Comments of Associate Editor (Dr. Cecilia Mascolo):

There are a few minor issues left which can be fixed with presentation adjustments. One aspect however to be discussed is related to ethics: one of the reviewers questions your choice of not putting this work through an ethics committee even just an overseeing body and we think this is probably best assessed by them before publication (there are ethics and legal aspects to be considered and possibly discussed in the paper).

Thank you very much for your contribution to the manuscript! We have provided an Ethic Review and detailed explanation for our study. We hope the revision could match the standard of publication in *Royal Society Open Science*.

Comments of Reviewer 1:

This paper collects information feeds by deploying 128 bots in Sina Weibo (chinese equivalent of Twitter). By analysing the text and network motifs in their social networks, the paper makes two discoveries/claims:

1. that filter bubbles have an “endogenetic” unidirectional star-like structure.
2. that the use of bots and AI could combines privacy protection and provides controllability in social media studies.

Thank you very much for your summary of the manuscript! Yes, we think the two points are the fundamental discoveries of the manuscript.

Related work is well introduced and placed in context. The paper itself is reasonably interesting, but I need more convincing arguments that it is valid, as explained below.

Thank you for your positive comments! We have improved your concerns in the revision, and hope these improvements will meet your requirements.

Main issue 1. The social bots not in the Random groups are designed to selectively enrich their connections within topic. So of course, we are bound to see “polarization” as seen in Section 3(a). In fact what is surprising is that there is not much polarization difference between STG and RSTG. To understand this difference, a complicated (and not entirely convincing) network analysis is performed. Instead, I imagine this arises just as a straightforward side effect of how much topic specialisation exists among the seed users that are followed, and in general all users in STG. What I imagine is that the users who are being followed by the bots in STG are users who tweet about science but also other topics ($R^0_{\text{STG}} = 0.5$). On the other hand, EG bots follow users who only tweet about entertainment and nothing else $R^0_{\text{EG}} = 76.77\%$. Therefore, when looking at the previous 50 messages after a random wakeup, and choosing users to follow, STG bots may find other topics, and users who tweet about those topics. In contrast, EG bots simply find other users also tweeting mainly about EG.

Good observation! Indeed, it may be confusing that the average initial (t=0) preferred topic ratio for EG (76.77%) is not equal to the ratio for STG (50%) in our experiment. The difference is because users that are preferring sci-tech topic generally have a lower proportion of posting or reposting the similar topic, usually around 40% to 60%. However, the ratio of users that are preferring entertainment topic can reach as high as over 80% in Weibo. That shows the entertainment topic is more polarized than the sci-tech topic initially. Nevertheless, the difference between EG and STG does not affect the analysis and results of our experiment. We have revised and explained in Section 3(a) with the following two reasons:

“Importantly, no matter how many different topics exposed to it after a random wake-up, the bots in topic groups only select a message consistent with its preference and follow the direct source of the selected

news. It means that the initial ($t=0$) preferred topic ratio does not affect the primary experimental process of choosing messages and following users of the bots. Finally, the initial difference between EG and STG provides a baseline for our further analysis.”

Main issue 2. There are only two topics tested. On one topic (EG) it is found (page 7, line 16) that selective exposure is necessary for polarization. On another topic (STG) this effect is not found. With this, the natural conclusion is that we simply cannot generalise, and that perhaps there is no universal truth being found - just some artefact of these specific topics.

Excellent suggestion! Yes, we just tested two topics and found the effect of selective exposure in the entertainment topic. Indeed, we cannot directly generalize the natural conclusion. Therefore, we have rewritten some inexact expressions in the manuscript. We have revised the sentence, “The result suggests that selective exposure is necessary for the polarization.” into “The result suggests that selective exposure is important but not sufficient for the polarization.”

Main issue 3. A third "cosmetic" issue is in the framing the claimed method - i.e. contribution 2: It is not clear why authors need to deploy “agents” or social bots. The same could have been achieved by simulating the following and reading policies of different kinds of users. At the heart is just the policy which is being followed, whether that is a bot, or a human. The fact that a bot has been used is “implementation detail”. I am also not sure how it avoids the "invasion of privacy" (Page 12, line 25).

Good point! In fact, this experiment is a Monte Carlo simulation analysis method running in a real social network environment. As for why we select deploying “agents” or social bots instead of employing volunteers, we revised the explanation in the first paragraph of the Conclusion section as: “(1) Compared with the experimental method based on volunteers\cite{burst18}, social bots use artificial intelligence technologies (e.g., natural language processing) to simulate specific

actions (e.g., selective exposure). The approach not only improves the freedom of the experimenters but also reduces the cost of the experiments on real-world social networks. Unlike volunteers, the behavior of social bots can be effectively controlled; therefore, the consistency and randomness of experiment processes can be ensured, such as activity frequency and content judgment criteria.”

As for your concern about privacy issues, we have added related explanation in the first paragraph of the Conclusion section: “(3) Compared with the experimental method of directly altering the real-world social network \cite{bond12}, this method can limit the interference to real-world social networks to a lower level. In our experiment, all social bots we deploy have no direct interaction with the existing users except following on them. The bots do not produce, forward, or modify any news content, and all data collected is directly exposed to them and publicly accessible. All network connections are restricted within bots and their direct followings, and we do not use the entire social relationship of all followings. That is, our experiment does not interfere with users’ behavior or disturb information dissemination, and all the data used is publicly visible. Therefore, the approach avoids the risk of violating the privacy of users.”

Meanwhile, given the concerns about ethics and privacy from you and reviewer 2, we have provided an ethical review material from the ethics committee organized by College of Computer Science and Technology, Zhejiang University of Technology. Finally, our experimental method is still in the exploration stage, and therefore, we also hope to hear more opinions through the publication of this manuscript.

Minor issue 1. “results from the algorithm are further voted...” —> was every step confirmed by three experimenters? How many steps did the bots take in total?

Good observation! Actually, the bots take five steps in total as expressed in the manuscript. We revised the step 3 to clarify the ambiguity: “In this step, we first run the FastText text classification algorithm

\cite{joulin16} to get preliminary classification results. To ensure the accuracy and correctness of classification, the results from the FastText algorithm were further voted by at least three experimenters. Although manual supervision is required, this algorithm can help us filter out a large number of inconsistent messages.”

Minor issue 2. Related to this, in line 32, it is stated that bots stop after reaching an upper limit of number of followers. It is not clarified until next page that this limit is based on Dunbar’s number. Please state here.

Thank you for your comments! We have moved the explanation of Dunbar number to the position. Now, the description of step 5 is: “If the following number reaches the upper limit, the bot stops running; otherwise, the bot becomes idle and waits to wake up again. The upper limit of followings for each bot is 150, according to the Dunbar number, which is a suggested cognitive limit to the number of people with whom one can maintain stable social relationships \cite{dunbar98}”. We hope these improvements will meet your requirements.

Minor issue 3. Page 7, line 29—30: "Compared with other topics, the two topics can be freely diffused in Weibo, have low overlap in content, and possess different user characteristics (gender, age, education level, etc.)." This needs a reference to confirm they have different user characteristics.

Good observation! We have added the two references (technical report from Weibo and Sina, the parent company of Weibo) to confirm they have different user characteristics.

Minor issue 4. Fig 2: Please do not use a pie chart. There are several statistical issues identified with pie charts (<https://www.data-to-viz.com/caveat/pie.html>)

Thank you for your suggestion! We have modified the figure, and

hope it will meet your requirements:

Minor issue 5. I don't quite follow the motifs explanation in section 3b. ii - it concludes saying STG networks contain more closed triangles with at least two reciprocal arcs. Isn't this to be expected given that dyadic analysis already found more reciprocity? Also, a couple of sentences are needed on how exactly this motif analysis explains the counterintuitive result noted in Page 10 line 1.

Good point! Previously, the explanation of counter-intuitive result is absent. In the revised manuscript, We have added a paragraph in the motif section to illustrate the result:

“The abundance of motifs 2D, 3F, and 3G confirm the counter-intuitive result derived from the statistical properties of the network, that is, the STG network is more clustering but less polarized. The form of motifs explains this result. In STG networks, if each peripheral node consumes only a small number of messages of other topics, the information has a relatively high probability of exposing to the central nodes. Consequently, the preference of central nodes may be affected, and broadcast the diverse messages to its followers, making various information more widely spread within the whole community.”

Comments of Reviewer 2:

The manuscript is a well written piece, that has a significant contribution to the field, and it is methodologically sound. The main strength of the study is that it is the first study that is comprehensively evaluating the use of social bot and artificial intelligence (although supervised) in researching inherent structure of filter bubbles occurring in online social spaces (in this case Weibo

microblogging network). It is also methodologically sound as author used four distinct groups: topic and random group, whereas researching filter bubble polarization processes/effects and network structures on two distinct topics: entertainment and sci-tech.

I could imagine that authors had difficulties singling-out and discussing few of many interesting findings their study yielded.

Thank you very much for your positive comment and valuable suggestions for the work! We have improved your concerns in the revision, and hope these improvements will meet your requirements.

Main issue 1. So, as results suggest, there is indeed difference between polarization potential and network structure between two given topics: entertainment and sci-tech. Network emerging within entertainment topic is more “unidirectional, star-like”, whereas sci-tech “bidirectional”. This is valuable finding and could be reflective to the topic/information per se – entertainment and sci-tech. Sci-tech network/community could be more “inductive” (being reflective and incorporating many diverse sources. And therefore, spreading diverse information), while entertainment “deductive” (being responsive to less diverse sources, ones dictated by few central agents). In other words, there is indeed difference in the content of information, and how it drives all the examined processes: polarization, self/other-reliance, network structure (although density seems to be similar), etc. So, I am a bit confused when conclusions of the manuscript (seen also in the abstract of the manuscript) are mostly based on findings that emerged around entertainment topic.

Moreover, as previously it has been suggested that similar issues indeed might occur in the “official” scientific community, and that similar processes might have impact on scientific evidence (even once considered as most reliable such as systematic reviews and meta-analyses). So, I would prefer to see more detailed discussion (within Discussion section) regarding the differences found within these two topics.

Excellent! Your suggestion is very instructive. As you pointed out, in the original manuscript we mainly use the results of the EG group for discussion, but there is a lack of discussion about STG. We have added a paragraph to further discuss the result about STG networks in section (iii) Visualization:

“Furthermore, our results may confirm the basic structure and behavior of the scientific community. The bidirectional scale-free structure of scientific collaboration networks has long been recognized. Our results show that the scientific community forms a similar interactive center-periphery structure, both in formal collaborations and in daily life (even in a unidirectional relationship-based social environment, e.g., microblogging). In Wu *et al.*, 2019, the researchers found an interactive pattern of large teams at the center of the scientific community and small teams at the periphery of the scientific community. In the pattern, small teams make disruptive innovation, while large teams further develop and spread such innovation. The pattern has a similar connotation to the bidirectional structure we found. Peripheric users in the scientific community are responsible for providing novel information, while the central user is responsible for further disseminating the information. As a result, the coupling of the scale-free structure and bidirectional information dissemination forms a positive feedback loop that stimulates scientific innovation. At the same time, the bidirectional structure of the scientific community can promote communication between users and limit the spread of misinformation compared to the star-like structure of the entertainment community.”

We hope these improvements will meet your requirements.

Technical issues 1. In collective and individual levels, the term ‘arpolarization’ as has two possible meanings. “– misspelling

Thank you for your correction! We have corrected the term in the manuscript.

Technical issues 2. In conclusion section points from (1) to (5) needs to be rearranged, as currently it is difficult to follow and there are some issues with spelling (for example after point 4).

Thank you for your suggestion! We have reconstructed the points and

corrected the spelling issues in conclusion section, and hope these improvements will meet your requirements:

“(1) Compared with the experimental method based on volunteer \cite{burst18}, social bots use artificial intelligence technologies (e.g., natural language processing) to simulate specific actions (e.g., selective exposure). The approach not only improves the freedom of the experimenters but also reduces the cost of the experiments on real-world social networks. Unlike volunteers, the behavior of social bots can be effectively controlled; therefore, the consistency and randomness of experiment processes can be ensured, such as activity frequency and content judgment criteria. (2) Compared with the experimental method based on artificial social networks \cite{centola10,centola11}, social bots can work directly on real-world social networks and acquire timely feedback with the instantaneous online environment. (3) Compared with the experimental method of directly altering the real-world social network \cite{bond12}, this method can limit the interference to real-world social networks to a lower level. In our experiment, all social bots we deploy have no direct interaction with the existing users except following on them. The bots do not produce, forward, or modify any news content, and all data collected is directly exposed to them and publicly accessible. All network connections are restricted within bots and their direct followings, and we do not use the entire social relationship of all followings. That is, our experiment does not interfere with users’ behavior or disturb information dissemination, and all the data used is publicly visible. Therefore, the approach avoids the risk of violating the privacy of users.”

Technical issues 3. Supplementary files can be easily accessed as mentioned within the manuscript, but website: www.socialbot.top is in Chinese.

We are very sorry about not taking this into account before! Now we have already translated the website into English.

Yours sincerely,

Yong Min

College of Computer Science and Technology,

Zhejiang University of Technology

Hangzhou 310058, P.R. China

Appendix B

Explanation Letter about Ethics Review of “Endogenetic structure of filter bubble in social networks”

Dear Editors and Reviewers,

Thanks to the editors and reviewers for their valuable comments on this article. For the ethical and legal issues involved in this experiment, I would like to make the following points:

First, in China, there are no official review mechanisms and related standards for sociology and economics research. Therefore, in the revision process, we entrusted our organization (College of Computer Science and Technology, Zhejiang University of Technology) to convene a temporary ethics review committee composed of third-party experts to review related issues.

Second, since it is impossible to predict which user the social bot will follow, and all activities occur in the Internet space, we cannot obtain the user's informed consent in advance. This is different from the usual experimental methods that rely on volunteer recruitment. Based on the design of the Weibo platform, each user will receive a notification when they are followed by our bots. A detail explanation is added to the robot account, and the user can view it freely.

Third, one of the innovations of this experimental solution is that it relies only on the limited activities of the robot in the real social media environment, and does not interfere with normal user activities and information dissemination processes. As the reviewer spends on the comments, this research method has a certain uniqueness. Therefore, we cannot find clear ethical standards to evaluate their impact. But we confirm that the activities of social robots have been limited to a minimum to complete the study.

Yours sincerely,

Yong Min

College of Computer Science and Technology,

Zhejiang University of Technology

Hangzhou 310058, P.R. China